# Medication Risks and Their Association with Patient-Reported Outcomes in Inpatients with Cancer

**DOI:** 10.3390/cancers16112110

**Published:** 2024-05-31

**Authors:** Maximilian Günther, Markus Schuler, Leopold Hentschel, Hanna Salm, Marie-Therese Schmitz, Ulrich Jaehde

**Affiliations:** 1Department of Clinical Pharmacy, Institute of Pharmacy, University of Bonn, 53121 Bonn, Germany; m.guenther@uni-bonn.de; 2Onkologischer Schwerpunkt am Oskar-Helene-Heim, 14195 Berlin, Germany; markus.schuler@onkologie-ohh.de; 3Division of Hematology, Oncology and Stem Cell Transplantation, Medical Clinic I, Department of Medicine I, University Hospital Carl Gustav Carus, Technical University of Dresden, 01307 Dresden, Germany; leopold.hentschel@ukdd.de; 4Klinik und Poliklinik für Innere Medizin C, Universitätsmedizin Greifswald, 17489 Greifswald, Germany; hanna.salm@stud.uni-greifswald.de; 5Sarcoma Center Berlin-Brandenburg, Helios Hospital Bad Saarow, 15526 Bad Saarow, Germany; 6Institute of Medical Biometry, Informatics and Epidemiology, Medical Faculty, University of Bonn, 53127 Bonn, Germany; m.schmitz@imbie.uni-bonn.de

**Keywords:** inpatient cancer care, patient-reported outcomes, medication risks, medication review, drug-related problems

## Abstract

**Simple Summary:**

Most cancer patients are older and have concomitant diseases because the incidence of most cancer types increases with age. This leads to patients taking a variety of medications that can cause drug-related problems (DRPs). DRPs can cause harm, including increased illness, avoidable hospital stays, and even death. Common DRPs are drug–drug interactions, not taking medication as prescribed, and adverse drug reactions. In our study, we aimed to assess these medication risks in hospitalized cancer patients and to identify factors that influence their health-related quality of life (HRQOL) as a patient-relevant outcome. The results of the pharmacist-led medication reviews show that DRPs are common in hospitalized cancer patients. Therefore, patient questionnaires about therapy-related symptoms could improve the detection of DRPs. While drug-related factors had no effect on HRQOL during the hospital stay, our analysis revealed other influencing factors, such as relapse status of the cancer disease and length of hospital stay.

**Abstract:**

Background: We aimed to assess medication risks and determine factors influencing the health-related quality of life (HRQOL) in cancer inpatients. Methods: A retrospective analysis was conducted to identify drug-related problems (DRPs) based on medication reviews, including patient-reported outcomes (PROs). Multiple linear regression analyses were performed to identify sociodemographic, disease-related, and drug therapy-related factors influencing changes from hospital admission to discharge in the scales of the EORTC QLQ-C30 questionnaire. Results: A total of 162 inpatients with various hematological and solid cancer diseases was analyzed. Patients received a mean of 11.6 drugs and 92.6% of patients exhibited polymedication resulting in a mean of 4.0 DRPs per patient. Based on PRO data, 21.5% of DRPs were identified. Multiple linear regression models described the variance of the changes in global HRQOL and physical function in a weak-to-moderate way. While drug therapy-related factors had no influence, relapse status and duration of hospital stay were identified as significant covariates for global HRQOL and physical function, respectively. Conclusion: This analysis describes underlying DRPs in a German cancer inpatient population. PROs provided valuable information for performing medication reviews. The multiple linear regression models for global HRQOL and physical function provided explanations for changes during hospital stay.

## 1. Introduction

Preventing medication risks in cancer patients represents a major challenge. Because the incidence of most tumor entities rises with age, most cancer patients are older and present comorbidity [1]. Compared to the general population, the comorbidity burden of cancer patients is higher [2]. With increasing comorbidity, the prevalence of polymedication also rises [3]. The common definition of polymedication is the use of five or more medications [4]. Hyperpolymedication is defined as the intake of 10 or more medicines [5]. Studies indicate that polymedication affects between 57% and 84% of older cancer patients and is associated with major risks [3,6,7,8,9,10]. Relations between polymedication and a range of health outcomes including adverse drug reactions (ADRs), falls, frailty, hospitalization, postoperative complications, and mortality were described [11,12,13,14].

One major reason for these negative outcomes upon polymedication are drug-related problems (DRPs), which are very common, especially in older cancer patients [15,16]. A DRP is defined as an event during pharmacotherapy which interferes with a desired health outcome [17]. The number of reported DRPs in cancer patients ranges between three and five per patient [6,18,19,20]. DRPs can lead to increased morbidity, unnecessary hospital admissions, and mortality [15,21]. Common DRPs are potential drug–drug interactions (DDIs), ADRs, and non-adherence [15,22]. A study on cancer patients found that 22.9% of admissions to the intensive care unit were associated with an ADR and the mortality rate of the admitted patients was 28.1% [23]. After being treated in a hospital, cancer patients are frequently readmitted to the hospital within 30 days and in almost 10% of cases, potentially because of a DRP [22].

In order to minimize the risks described for cancer patients, medication management was developed, a co-operation of all healthcare professionals involved in the medication process. The patient’s overall medication regimen is analyzed for DRPs (medication review), followed by multiprofessional care of the patient focusing on predefined treatment goals. According to the available information, three types of medication reviews can be defined: simple medication review (type 1), advanced medication review (type 2a or 2b), and complete medication review (type 3). The simple medication review (type 1) is based on medication data and basic patient data. The advanced medication review is additionally supported by a patient interview (type 2a) or clinical data, including diagnoses and laboratory parameters (type 2b). The complete medication review (type 3) is based on all the abovementioned sources of information [24].

The effect of pharmacist interventions on adult outpatients with cancer was summarized in several systematic reviews with the result that the interventions could improve outcome measures like rates of nausea, vomiting and pain control, medication adherence, patient satisfaction, quality of life, and cost savings [25,26,27,28]. A positive effect on the number of DRPs, clinical outcomes, and care processes was also shown in several studies [6,18,29].

Healthcare professionals can gain important information about their patients by assessing patient-reported outcomes (PROs) [30]. Systematic monitoring of PROs is associated with improved patient–clinician communication, clinician awareness of symptoms, symptom management, and patient satisfaction [31,32,33,34,35]. Furthermore, patient-relevant outcomes such as health-related quality of life (HRQOL) and overall survival can be improved [35,36]. Therefore, it is crucial to incorporate PROs into health care interventions, such as medication reviews, and consider HRQOL as a patient-relevant outcome for evaluation.

Most research in the field of medication safety in oncology was conducted in an outpatient setting and focuses on long-term effects of pharmaceutical care interventions. For inpatient cancer care, information about frequent DRPs and the effect of medication reviews and medication management is still limited. It is conceivable that inpatient cancer patients are exposed to distinct medication risks and that pharmaceutical interventions are associated with different challenges compared to the outpatient setting.

With this analysis, we aimed to assess medication risks in cancer inpatients and determine sociodemographic, disease-related, and drug therapy-related factors influencing HRQOL in oncology inpatients. A scheme of the study framework is shown in Figure 1.

## 2. Materials and Methods

### 2.1. Study Design

This project was a post-hoc conducted, but in advance planned, retrospective analysis including cancer inpatients of the database created during a randomized controlled trial conducted at four oncology wards of Helios hospitals in Berlin between July 2017 and February 2019 [37]. The primary aim of the underlying study was to evaluate the feasibility of an electronic patient-reported outcome (ePRO) assessment in inpatient cancer care [37]. The patients were assigned to three groups: intervention group A (ePRO assessment with presentation of patient answers to physicians and the opportunity to adapt the therapy in response), control group B (ePRO assessment without presentation of patient answers to physicians), and control group C (paper-based PRO assessment without presentation of patient answers to physicians).

In this retrospective secondary patient data analysis, medication risks were detected with the help of medication reviews, including PRO data. Subsequently, sociodemographic, disease-related, and drug therapy-related factors influencing changes in the different dimensions of HRQOL from hospital admission to discharge were determined.

The Ethics Committee of the Chamber of Physicians in Berlin, Germany, approved the study (Eth-48/16). It was carried out in accordance with the applicable German and European legal provisions as well as the Declaration of Helsinki [38].

Patients with hematological or oncological cancer entities of at least 18 years were included in the study. The planned inpatient stay had to be at least three days. Investigators recruited patients on the day of inpatient admission. Patients were informed about the study and signed a written informed consent. Patients with no or insufficient documentation of their medication within the study database were excluded from the secondary analysis. All data were anonymized before analysis.

### 2.2. Patient Documentation

For this secondary analysis, relevant patient data were extracted from the study database after the end of the study. The time-points of documentation are shown in Figure 2.

Baseline documentation was undertaken at hospital admission. The following data were documented: eligibility criteria, sociodemographic parameters, and disease-anamnestic data on cancer diagnosis and concomitant diseases.

The following data were documented independent of time during the hospital stay: current therapeutic regimen and tumor therapy, concomitant medication for comorbidity, supportive drug therapy, laboratory data, and vital parameters.

### 2.3. Patient-Reported Outcomes

At baseline, visit 1, visit 2, and visit 3, symptom burden and health-related quality of life were assessed as ePROs via a tablet-based digital solution (group A and B) or paper-based (group C).

The symptom burden was evaluated using the PRO version of the Common Terminology Criteria for Adverse Events (PRO-CTCAE). The PRO-CTCAE item library was developed as a complementary tool to the CTCAE criteria by the US National Cancer Institute [39]. It consists of 124 items representing 78 symptomatic adverse events. The items characterize up to three symptom attributes regarding severity, frequency, and interference with daily activities [40]. Answers are requested on a verbal five-point Likert scale and refer to the last seven days. A German translation is available [41]. In this study, a validated German PRO-CTCAE core item set for patients with chemotherapy containing 31 items was used [42]. PRO-CTCAE symptom scores were calculated using the following equations. First, the raw score (RS), indicating the mean value of the symptom attributes, was calculated with Equation (1) if at least 50% of attributes were answered [43]:(1)$$\mathrm{RS}=\frac{I_{1}{+I}_{2}+\ldots {+I}_{n}}{n}$$
I_1_ = Value of item 1;I_2_ = Value of item 2;I_n_ = Value of item n;n = Number of items per scale.


Second, the raw score was linearly transformed to numerical score values ranging from 0 to 100 using Equation (2) [34]; higher values indicate a higher severity of the symptom:(2)$$\mathrm{Score}=\left\{\frac{\mathrm{RS}}{\mathrm{Range}}\right\}\times 100$$
Range = Difference between maximum and minimum values of the response scale (0 to 4).

HRQOL was assessed using the Quality-of-Life Questionnaire-Core 30 (QLQ-C30) Version 3.0 of the European Organization for Research and Treatment of Cancer (EORTC) [44]. The questionnaire is validated and a certified German translation is available [44,45]. It encompasses the global HRQOL scale, five functional subscales, and nine symptom scales, with a total of 30 items. Scoring was performed using the EORTC QLQ-C30 scoring manual [43]. Higher scores on the functioning subscales and the global HRQOL indicate better outcomes, whereas higher values on the symptom scales indicate a higher symptom burden.

### 2.4. Analysis of Medication Risks

Medication risks were recorded as DRPs. The Pharmaceutical Care Network Europe Association (PCNE) defines a DRP as follows: “A Drug-Related Problem is an event or circumstance involving drug therapy that actually or potentially interferes with desired health outcomes” [17]. According to PCNE, a medication review is a “structured evaluation of a patient‘s medicines with the aim of optimising medicines use and improving health outcomes. This entails detecting drug-related problems and recommending interventions” [46]. Based on medication and clinical data, retrospective advanced medication reviews of type 2b were conducted to identify DRPs [24]. Some DRPs such as administration problems and non-adherence can only be detected in a complete medication review type 3, for which additional patient interviews would be mandatory. Instead, patients reported their symptoms under therapy using the PRO-CTCAE questionnaires. In this analysis, this source of information was used to complete the medication reviews in the absence of patient interviews to detect a broader spectrum of DRPs. The DRPs identified by PRO-CTCAE are henceforth referred to as PRO-DRPs in the following. Because therapy-related symptoms are an inherent part of tumor therapy, only a severe symptom burden with a PRO-CTCAE symptom score of at least 75 was considered as a DRP.

The DRP categories considered in the medication reviews are shown in Table 1.

In general, a DRP was only recorded if a pharmaceutical intervention, as indicated by a therapy recommendation, would have been required [47].

### 2.5. Statistical Analysis

For data entry and the processing and statistical analysis of the anonymized data, Microsoft Office^®^ Professional Plus 2019 (Microsoft Corporation, Redmond, WA, USA), IBM SPSS^®^ Statistics 27.0 (IBM Corporation, Armonk, NY, USA), and R 4.0.5 (The R Foundation for Statistical Computing, Vienna, Austria) were used.

Descriptive statistics were performed for patient characteristics, medication data, DRPs, and PROs. Mean values with standard deviations (SDs) or the median with interquartile range (IQR) were calculated, as applicable. Frequencies were described as absolute numbers and percentages [48,49]. In the exploratory analysis, a *p*-value of <0.05 was considered as statistically significant. Confidence intervals (CIs) of 95% were calculated.

Associations between changes in global HRQOL and the subscales physical function, cognitive function, and emotional function of the EORTC QLQ-C30 questionnaire from baseline to hospital discharge (visit 2) with several parameters were investigated by multiple linear regression models. The following prespecified independent variables were included in the exploratory regression analysis: study group (A, B, C), age (years), gender (male, female), educational level (high, low), hospital stay (days), cancer type (solid, hematological), time since cancer diagnosis (months), relapse status (yes, no), ECOG status (0, 1, 2, 3), concomitant diseases (number), drugs (number), DRPs (number), and PRO-DRPs (number). The goodness-of-fit of the multiple linear regression model was evaluated by using R^2^ and adjusted R^2^. It can be interpreted according to Cohen [50].

## 3. Results

### 3.1. Study Population

In total, 185 patients were included in the underlying clinical trial. Of these, 18 patients dropped out of the study. Five patients had to be excluded from the secondary data analysis because of missing documentation in the study database. Thus, the target sample for the secondary data analysis resulted in 162 patients.

The median age of the patients was 65.5 years (IQR: 18, range: 19–88) and 86 patients (53.1%) were at least 65 years old. The proportion of female patients was 56.2% (n = 91). The median hospital stay was four days (IQR: 7, range: 1–73). Most patients (n = 103, 63.6%) had a solid tumor disease. Tumor entities presented in at least 5% of patients were lymphoma (n = 31, 19.1%), sarcoma (n = 20, 12.3%), rectal cancer (n = 16, 9.9%), leukemia (n = 15, 9.3%), multiple myeloma (n = 12, 7.4%), esophageal cancer (n = 11, 6.8%), pancreatic cancer (n = 10, 6.2%), and colon cancer (n = 9, 5.6%). No relapse of their tumor disease occurred in 123 patients (75.9%). Four months represented the median time since the first diagnosis of cancer (IQR: 13.5, range: 0–208). According to their ECOG status, most patients had a rather good physical condition reflected by 87 patients (53.8%) with an ECOG status of 0 and 53 patients (32.7%) with an ECOG status of 1. The mean number of concomitant diseases listed in the Charlson Comorbidity Index (CCI) was 0.97 (SD: 1.25, range 0–6). Concomitant diseases present in at least 5% of patients were mild/severe kidney disease (18.5%), chronic pulmonary disease (13.6%), secondary tumor disease (11.1%), secondary metastatic tumor disease (10.5%), diabetes mellitus with organ damage (10.5%), diabetes mellitus without organ damage (8.0%), and heart failure (8.0%). A mean of 11.6 drugs per patient (SD: 5.15, range: 2–26, median: 11, IQR: 7), including cancer treatment and supportive/concomitant medication, was administered. Patients with solid tumor diseases (mean: 12.8, SD: 5.08, range: 3–26, median: 12, IQR: 7) received a higher number of drugs than patients with hematological diseases (mean: 9.7, SD: 4.54, range: 2–21, median: 9, IQR: 8). In total, 1884 drugs were administered to patients. Figure 3 shows the drug classes used according to their Anatomical Therapeutical Chemical (ATC) code level 1. Appendix A shows the drugs according to ATC code level 2 as well. ATC drug classes level 2 with more than 5% of drugs used were in group A (alimentary system and metabolism) remedies for acid-related diseases (A02; n = 113, 6.0%) and antiemetics/anti-nausea agents (A04; n = 168, 8.9%), in group B (blood and hematopoietic organs) antithrombotic agents (B01; n = 100, 5.3%), in group H (systemic hormone preparations excluding sexual hormones and insulin) corticosteroids for systematic use (H02; n = 136, 7.2%), in group L (antineoplastic and immunomodulatory agents) antineoplastic agents (L01; n = 372, 19.7%), and in group N (nervous system) analgesics (N02, n = 111, 5.9%). Polymedication with five or more drugs occurred in 150 patients (92.6%) during their hospital stay, and 98 patients (60.5%) exhibited hyperpolymedication with 10 or more drugs.

### 3.2. Drug-Related Problems

The mean number of DRPs was 4.0 per patient (SD: 2.97, range: 0–13, median: 3.0, IQR: 4) during the hospital stay. Patients with solid tumor diseases (mean: 3.2, SD: 2.17, range: 0–10, median: 3, IQR: 4) experienced a lower number of DRPs than patients with hematological diseases (mean: 5.3, SD: 3.51, range: 0–13, median: 4, IQR: 5). Table 2 shows the number of DRPs per category. The three DRP categories indication without drug, inappropriate drug choice, and adverse drug reaction, are among the five most common DRP categories across the three patient groups (all patients, patients with solid tumor diseases, and patients with hematological tumor diseases).

Of the 641 in total detected DRPs, 138 DRPs (21.5%) could be detected by a PRO-CTCAE symptom score of at least 75 (PRO-DRP). At the time-points baseline and visit 2 (hospital discharge), the number of severe symptoms per patient was calculated, as these questionnaires were administered to every patient in the study. At baseline, 1.47 (SD: 2.03, range: 0–12, median: 1, IQR: 2) severe symptoms with a score of at least 75 occurred per patient. At visit 2, the number was 1.58 (SD: 2.32, range: 0–10, median: 1, IQR: 2). Across all visits, fatigue (26.5%) was the most often-occurring severe patient-reported symptom, followed by decreased appetite (17.4%) and insomnia (15.3%).

### 3.3. Determinants of Health-Related Quality of Life

The changes in the scales could not be calculated in 5.6% (n = 9) of cases for the global HRQOL and cognitive and emotional function, and in 6.2% (n = 10) of cases for the physical function due to missing values. The distribution of the changes in global HRQOL and physical function from baseline to hospital discharge is shown in the histograms of Figure 4. For histograms of cognitive function and emotional function, see Appendix A.

The results of the multiple linear regression models on the change in global HRQOL and the physical function subscale of the EORTC QLQ-C30 questionnaire are shown in Table 3 and Table 4.

The model for global HRQOL described a significant variance of the change in the global HRQOL scale (*p* = 0.031), but with a weak-to-moderate variance explanation: the included independent variables explained 8.6% of the variance of the dependent variable (R^2^ = 0.184, adjusted R^2^ = 0.086). While drug therapy-related factors (“Drugs”, “DRPs”, and “PRO-DRPs”) had no influence, the variable “Relapse status” significantly influenced the change in global HRQOL (*p* = 0.013). Patients without a current relapse of their tumor disease showed on average an increase of 11.06 points on the global HRQOL scale of the EORTC QLQ-C30 questionnaire from baseline to hospital discharge.

The model for physical function described the variance significantly (*p* = 0.009) and with a weak-to-moderate variance explanation as well. The independent variables explained 11.6% of the variance of the dependent variable (R^2^ = 0.211, adjusted R^2^ = 0.116). While drug therapy-related factors (“Drugs”, “DRPs”, and “PRO-DRPs”) had no influence, “Hospital stay” had a significant influence on the change in physical function (*p* = 0.009). With every additional day that the patients stayed in hospital, the physical function scale decreased on average by 0.48 points.

The models for cognitive function (*p* = 0.122) and emotional function (*p* = 0.210) did not describe a significant variance of the changes in the scales.

## 4. Discussion

With this analysis, we aimed to assess medication risks and determine factors influencing the HRQOL in oncology inpatients. The results can contribute to the development of supportive care concepts for cancer inpatients.

The underlying clinical trial was conducted to evaluate the feasibility of a multidimensional electronic PRO (ePRO) system and to implement it in an inpatient oncology setting. The primary analysis indicates that the ePRO tool is feasible, but symptom burden and global HRQOL did not change significantly between intervention group A and control group B. These findings suggest that physicians did not respond adequately to the ePRO measures because there was no predefined supportive care concept for the study [37].

### 4.1. Medication Risks

The patients received a mean of 11.6 drugs per patient, including cancer treatment, supportive, and concomitant medication. This number includes drugs used only for a short duration (e.g., antiemetic prophylaxis). Because drugs were documented independent of time at discharge, not all drugs were necessarily administered concurrently and no differentiation in drugs before and after hospital admission was possible. This approach may overestimate the number of drugs. Nevertheless, every administered drug can cause a DRP. The number of drugs per patient corresponds to the findings of most comparable studies [6,18,19], which is also true for the number of patients with polymedication. In contrast, the number of patients with hyperpolymedication is slightly higher in this study [3,6,7]. Furthermore, a recent prospective study by Lavan et al. on older cancer out- and inpatients observed lower numbers of drugs than our study. A median of seven drugs was regularly prescribed to patients, resulting in 61% of patients with polymedication and 18% of patients with hyperpolymedication [10]. The difference may reflect the more frequent prescription of drugs in hospitals in addition to the regularly used drugs and emphasizes the drug-related risks of cancer inpatients.

DRPs amounted to 4.0 DRPs per patient. This number was lower, compared to a retrospective study by Vucur et al. on head and neck cancer outpatients. DRPs per patient ranged from 4.8 (first therapy cycle) to 6.9 (fifth therapy cycle) [19]. However, our findings correspond to the studies of Nightingale et al. and Tan et al., with three DRPs per patient, and Edwards et al., showing 3.7 DRPs per patient [6,18,20]. Interestingly, patients with solid tumor diseases received more drugs per patient (12.8 vs. 9.7, respectively) but experienced fewer drug-related problems (3.2 vs. 5.3, respectively). The reason for this result is unclear and deserves further investigation.

In our study with cancer inpatients, the most frequently encountered categories of DRPs were indication without drug, inappropriate drug choice, and adverse drug reactions. Even across the subgroups of patients with solid and hematological tumor diseases, these three symptoms ranged within the five most common DRPs. In comparison, in studies with an outpatient setting, the following heterogeneous DRP categories occurred most often: indication without drug, adverse drug reactions, drug–drug interactions, inappropriate duration of use, drug without indication, inappropriate dosage, and non-adherence [6,15,18,19]. A study by Umar et al. on Turkish cancer inpatients reported adverse drug reactions, indication without drug, and drug without indication as most frequent DRP categories [29]. Non-adherence could not be detected in the present study, but it is likely that outpatients are more often affected because they are responsible for taking their own medication.

A limitation of this study is that medication reviews were only conducted retrospectively and therefore interventions could not be undertaken. The substantial proportion of DRPs that are associated with PRO symptoms (21.5%) indicates that including PRO data in medication reviews improves the detection of medication risks in cancer inpatients.

### 4.2. Determinants of Health-Related Quality of Life

The minor changes in scores, that can be considered clinically meaningful for patients, are called minimal important differences (MIDs). For the EORTC QLQ-C30 questionnaire, the MIDs for the global HRQOL and its subscales were evaluated in different studies [51,52,53]. As an approximation, values between five and ten can be assumed small patient-relevant changes. Values between 10 and 20 indicate a moderate difference, and values above 20 indicate a significant difference [54,55]. Therefore, it is possible to consider that all patients distributed to bin centers ±10 or higher experienced patient-relevant changes in their HRQOL (see Figure 4).

For the global HRQOL and physical function, we found multiple linear regression models describing the variance of the changes from baseline to hospital discharge. The models for cognitive function and emotional function were not significant. A possible explanation is that cognitive function and emotional function are more complex constructs of HRQOL, that require a longer time period to change than, for example, physical function. Therefore, the sample size of the study population may have been insufficient and the duration of the hospital stay too brief to manifest a significant change in these subscales of HRQOL. The models for global HRQOL and physical function may have been affected as well, but they still describe the changes in the dependent variables in a weak-to-moderate way [50].

The above circumstances may also account for the fact that there was no association with changes in HRQOL for the drug therapy-related factors “Drugs”, “DRPs”, and “PRO-DRPs”. In particular, for “PRO-DRPs”, an association with HRQOL would have been plausible because they are largely represented by symptomatic adverse events. Furthermore, the impact of pharmaceutical interventions resolving DRPs on symptom control and HRQOL has been shown [26,28]. However, because of the retrospective character of the medication reviews, no interventions to resolve DRPs and thereby influence the change in HRQOL during hospital stay could be carried out in this study.

Within the model for global HRQOL, only the variable “Relapse status” had a significant influence on the change in global HRQOL. This finding aligns with the expectation that patients with no relapse of their cancer disease benefit the most from the treatment during their hospital stay. For most cancer types, the treatment possibilities are best in the early stages of the disease and decrease upon relapses. The increase of 11.1 points on the global HRQOL scale is above the MID value and indicates a moderate patient-relevant difference [51,52].

Regarding the model for physical function, “Hospital stay” had a negatively directed influence on the change in physical function. The deterioration of physical function per day spent in hospital is not unexpected. The inpatient setting itself and therapy-related adverse events such as symptomatic and hematological toxicity like myelosuppression triggered by many chemotherapeutics affect the patients’ physical function during the hospital stay, especially if they appear closely after treatment.

### 4.3. Approaches to Improve Medication Safety

One approach to improve medication safety of cancer patients is the implementation of best-practice models addressing risk-adapted and interprofessional supportive care. In healthcare, best practice is defined as the optimization of a care process using evidence-based decision making for patients to ensure ongoing quality assurance [56]. ePRO assessment, pharmacist-led medication reviews, and other components like psycho-oncological care or therapeutic drug monitoring can be combined to improve patients’ health outcomes. Such complex interventions should be developed and evaluated in a phased approach. In the first phase, components of the intervention must be identified [57]. Thereafter, the following questions need to be answered: What is the trigger for an intervention? (e.g., PRO-CTCAE symptom over defined cut-off value, polymedication); how is the intervention conducted? (e.g., performing a symptom management, type of medication review performed and implementation of solutions); and which patient-relevant outcomes should be measured? (e.g., HRQOL, therapy discontinuation, adherence to treatment). Subsequently, the effectiveness of the complex intervention should be evaluated in exploratory and randomized controlled trials.

The results of this analysis can serve as a basis for designing supportive care concepts for German cancer inpatients, which are not yet established in clinical routine. In particular, pharmacist-led medication reviews, patient-reported symptom, and HRQOL monitoring are not yet routine clinical practices for German cancer inpatients. Medication reviews seem to be an important component, because most patients exhibited polymedication or even hyperpolymedication. The results show which categories of DRPs are of particular interest for the inpatient population. Moreover, PRO symptoms provided important additional information, which may increase the outcome of such medication reviews.

## 5. Conclusions

This secondary analysis of a clinical trial describes underlying DRPs in a German cancer inpatient population. Data on DRPs in cancer inpatients are limited because the majority of studies on pharmaceutical care are conducted on cancer outpatients. The patient population of the current trial showed a high proportion of patients with polymedication or even hyperpolymedication, indicating that these patients benefit from medication reviews. PRO symptoms provided important additional information, which may increase the outcome of such medication reviews.

The evaluation of the HRQOL suggests that a substantial proportion of patients underwent relevant changes in their HRQOL during hospital stay. While drug therapy-related factors had no influence, the multiple linear regression models for the global HRQOL and the physical function of the EORTC QLQ-C30 questionnaire partly explain the changes in HRQOL of the study population.

## Figures and Tables

**Figure 1 cancers-16-02110-f001:**
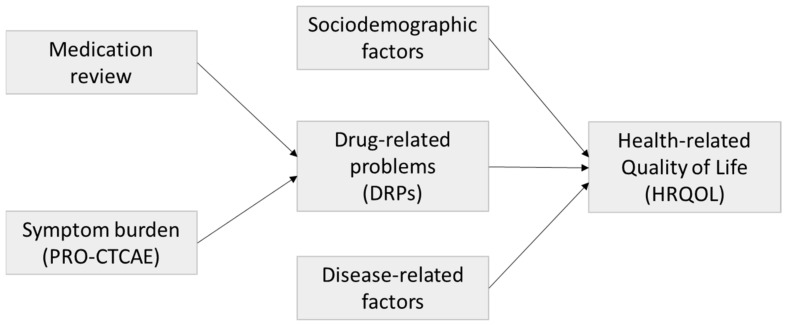
Study framework.

**Figure 2 cancers-16-02110-f002:**
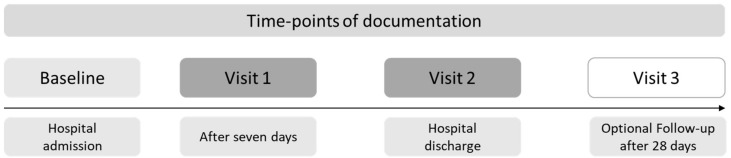
Time-points of documentation of the clinical trial. Visit 1 was only performed for patients with a hospital stay of at least seven days.

**Figure 3 cancers-16-02110-f003:**
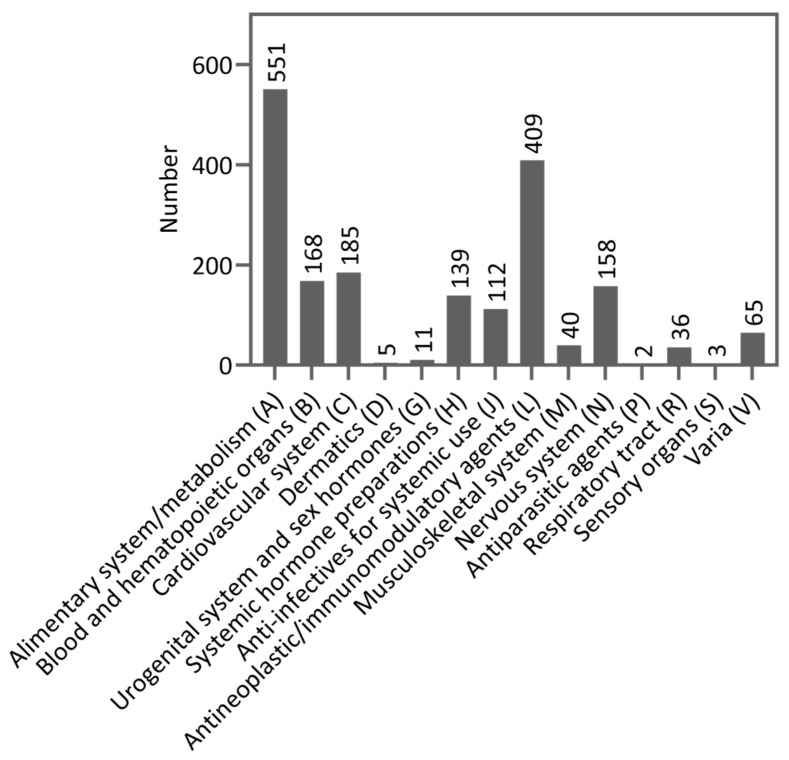
Drug classes categorized by Anatomical Therapeutical Chemical (ATC) code level 1 (n = 1884).

**Figure 4 cancers-16-02110-f004:**
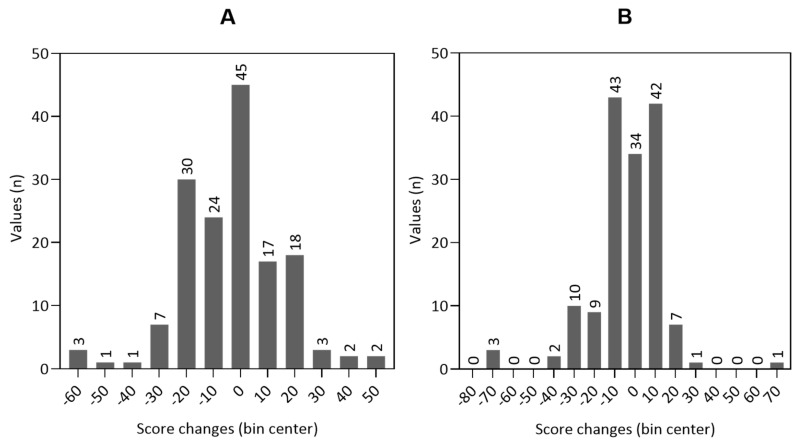
Histograms of the distribution of the score changes from baseline to hospital discharge in global health-related quality of life ((**A**) n = 153) and physical function ((**B**) n = 152).

**Table 1 cancers-16-02110-t001:** Categories of drug-related problems (DRPs) in medication reviews of type 2b [24].

Administration problem *
Adverse drug reaction
Contraindication due to age and gender
Contraindication due to comorbidity
Double medication
Drug without indication
Drug–drug interaction
Drug–food interaction *
Inappropriate administration time
Inappropriate dosage
Inappropriate dosage form #
Inappropriate dosage interval
Inappropriate drug choice
Inappropriate duration of use
Indication without drug
Non-adherence *

* Not detectable in medication reviews of type 2b. # Only partially detectable in medication reviews of type 2b.

**Table 2 cancers-16-02110-t002:** Number of drug-related problems (DRPs) per category for all patients (n = 162) and subgroups of patients with solid (n = 103) and hematological (n = 58) tumor diseases.

DRP Category	Number of DRPs (%) All Patients	Number of DRPs (%) Solid	Number of DRPs (%) Hematological
Adverse drug reaction	50 (7.8)	20 (6.1)	28 (9.1)
Contraindication due to age and gender	0 (0)	0 (0)	0 (0)
Contraindication due to comorbidity	3 (0.5)	1 (0.3)	2 (0.7)
Double medication	6 (0.9)	6 (1.8)	0 (0)
Drug without indication	41 (6.4)	33 (10.1)	8 (2.6)
Drug–drug interaction	34 (5.3)	29 (8.9)	5 (1.6)
Inappropriate administration time	19 (3.0)	9 (2.8)	10 (3.3)
Inappropriate dosage	32 (5.0)	19 (5.8)	13 (4.2)
Inappropriate dosage form	1 (0.2)	1 (0.3)	0 (0)
Inappropriate dosage interval	44 (6.7)	17 (5.2)	25 (8.1)
Inappropriate drug choice	83 (12.9)	68 (20.9)	15 (4.9)
Inappropriate duration of use	14 (2.1)	12 (3.7)	2 (0.7)
Indication without drug	314 (49.0)	111 (34.1)	199 (64.8)
Total	641 (100)	326 (100)	307 (100)

Subgroup allocation was not possible for one patient with DRPs: adverse drug reaction (n = 2), inappropriate dosage interval (n = 2), and indication without drug (n = 4).

**Table 3 cancers-16-02110-t003:** Results of the multiple linear regression analysis on the change in the global HRQOL scale of the EORTC QLQ-C30 questionnaire from baseline to hospital discharge.

Independent Variable	Estimate	SE	*p*	95% CI	
Study group B	−4.85	3.77	0.202	−12.31	2.62
Study group C	−6.20	3.54	0.083	−13.22	0.81
Age (years)	0.23	0.13	0.088	−0.04	0.49
Gender (female)	0.76	3.21	0.814	−5.59	7.11
Educational level (low)	1.44	3.02	0.635	−4.54	7.41
Hospital stay (days)	−0.34	0.19	0.082	−0.71	0.04
Cancer type (solid)	−0.29	3.87	0.940	−7.94	7.36
Time since cancer diagnosis (months)	0.04	0.05	0.415	−0.06	0.15
Relapse status (no)	11.06	4.38	0.013 *	2.38	19.73
ECOG status 1	2.83	3.55	0.426	−4.19	9.85
ECOG status 2	10.85	5.91	0.069	−0.85	22.54
Concomitant diseases (number)	−1.32	1.41	0.350	−4.12	1.47
Drugs (number)	0.18	0.36	0.613	−0.53	0.89
DRPs (number)	−0.10	0.72	0.896	−1.53	1.34
PRO-DRPs (number)	−0.30	1.43	0.833	−3.13	2.53

SE = standard error, CI = confidence interval, ECOG = Eastern Cooperative Oncology Group, DRP = drug-related problem, PRO = patient-reported outcome, * significant result.

**Table 4 cancers-16-02110-t004:** Results of the multiple linear regression analysis on the change in the physical function scale of the EORTC QLQ-C30 questionnaire from baseline to hospital discharge.

Independent Variable	Estimate	SE	*p*	95% CI	
Study group B	−6.08	3.59	0.093	−13.19	1.03
Study group C	−4.44	3.37	0.190	−11.12	2.23
Age (years)	0.15	0.13	0.245	−0.10	0.40
Gender (female)	−0.17	3.04	0.957	−6.17	5.84
Educational level (low)	−3.11	2.86	0.279	−8.77	2.55
Hospital stay (days)	−0.48	0.18	0.009 *	−0.84	−0.12
Cancer type (solid)	1.79	3.71	0.630	−5.55	9.14
Time since cancer diagnosis (months)	0.05	0.05	0.299	−0.05	0.15
Relapse status (no)	1.49	4.14	0.720	−6.71	9.68
ECOG status 1	1.17	3.36	0.728	−5.47	7.81
ECOG status 2	4.33	5.61	0.441	−6.76	15.42
Concomitant diseases (number)	−0.83	1.34	0.540	−3.48	1.83
Drugs (number)	0.52	0.34	0.135	−0.16	1.20
DRPs (number)	−0.05	0.69	0.942	−1.41	1.31
PRO-DRPs (number)	−0.88	1.35	0.516	−3.55	1.79

SE = standard error, CI = confidence interval, ECOG = Eastern Cooperative Oncology Group, DRP = drug-related problem, PRO = patient-reported outcome, * significant result.

## Data Availability

The datasets generated during and/or analyzed during the current study are available from the corresponding author on reasonable request.

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
