# Peer review of "Medication Risks and Their Association with Patient-Reported Outcomes in Inpatients with Cancer"

_cancers, 2024, doi:10.3390/cancers16112110_

Round 1

Reviewer 1 Report

Comments and Suggestions for Authors

The manuscript “Medication risks and their association with patient-reported outcomes in inpatients with cancer” is devoted to the assessment of various risks when taking chemotherapy along with the routine use of various medications in cancer patients with comorbid diseases. The article is written in literate literary language and contains a large amount of data on the general condition of the patients, but not histological diagnoses. Also, data on patients is presented quite briefly and generally without dividing into stages. Outcomes, comorbidity, and the spectrum of diseases are presented very superficially.

Therefore, I believe that this manuscript is more suitable for publication in Nursing Reports.

Reviewer 2 Report

Comments and Suggestions for Authors

This paper Medication risks and their association with patient-reported 2 outcomes in inpatients with cancer is well prepared, however I have a few comments.

1 .  Please add 1 figure to introduction with general idea of the paper

2. Material and methods section is described with details but corresponding to this results section is not enough detailed . Please consider to improve.

3. Please consider add some photo and 1 more figure to the results section.

4. Discussion is very limited with references

5. References are at least 10 years old, please add something current and please include in discussion.

 Thank you!

Reviewer 3 Report

Comments and Suggestions for Authors

This study aimed to explore medication risks, specifically drug-related problems, among cancer patients. It conducted a retrospective analysis involving 162 hospitalized cancer patients. Upon examination, it was observed that while the focus on medication risks in cancer patients is pertinent, the study lacks specificity. The authors intended to investigate drug-related problems but failed to specify the types of medication involved. Additionally, they did not differentiate between patients with hematological and solid tumors, which is crucial for a comprehensive analysis. Moreover, essential patient data such as gender, age, cancer type, and stage were not adequately categorized and should be integrated into the research model.

Comments on the Quality of English Language

The English is alright to read.

Round 2

Reviewer 1 Report

Comments and Suggestions for Authors

I am completely satisfied with the changes made.

Reviewer 2 Report

Comments and Suggestions for Authors

Thank you

Reviewer 3 Report

Comments and Suggestions for Authors

I accept the modifications from the authors as per my comments.

Comments on the Quality of English Language

No comment.